# Effect of Central Obesity and Hyperandrogenism on Selected Inflammatory Markers in Patients with PCOS: A WHtR-Matched Case-Control Study

**DOI:** 10.3390/jcm9093024

**Published:** 2020-09-20

**Authors:** Małgorzata Kałużna, Magdalena Człapka-Matyasik, Katarzyna Wachowiak-Ochmańska, Jerzy Moczko, Jolanta Kaczmarek, Adam Janicki, Katarzyna Piątek, Marek Ruchała, Katarzyna Ziemnicka

**Affiliations:** 1Department of Endocrinology, Metabolism and Internal Diseases, Poznan University of Medical Sciences, 49 Przybyszewskiego St, 60-355 Poznań, Poland; a.janicki18@gmail.com (A.J.); katarzyna.piatek.kp@ump.edu.pl (K.P.); mruchala@ump.edu.pl (M.R.); kaziem@ump.edu.pl (K.Z.); 2Department of Human Nutrition and Dietetics, Poznan University of Life Sciences, Wojska Polskiego 28, 60-637 Poznan, Poland; magdalena.matyasik@up.poznan.pl; 3Ward of Endocrinology, Metabolism and Internal Diseases Ward, Heliodor Święcicki Clinical Hospital, 49 Przybyszewskiego St, 60-355 Poznań, Poland; wachowiakkat@gmail.com; 4Department of Computer Science and Statistics, Poznan University of Medical Sciences, 7 Rokietnicka St, 60-806 Poznań, Poland; jmoczko@ump.edu.pl; 5Central Laboratory, Heliodor Swiecicki University Hospital, 49 Przybyszewskiego St, 60-355 Poznań, Poland; kaczmarek.jolanta@spsk2.pl

**Keywords:** polycystic ovary syndrome (PCOS), systemic inflammation, waist-to-height ratio (WHtR), white blood cell counts (WBC), lymphocyte-to-monocyte ratio (LMR), monocyte-to-high-density lipoprotein cholesterol ratio (MHR), high-sensitivity C-reactive protein (hsCRP), hyperandrogenism, insulin resistance (IR), metabolic syndrome (MS)

## Abstract

White blood cell counts (WBC), lymphocyte-to-monocyte ratio (LMR), and monocyte-to-high-density lipoprotein cholesterol ratio (MHR) are used as chronic inflammation markers. Polycystic ovary syndrome (PCOS) is a constellation of systemic inflammation linked to central obesity (CO), hyperandrogenism, insulin resistance, and metabolic syndrome. The waist-to-height ratio (WHtR) constitutes a highest-concordance anthropometric CO measure. This study aims to access WBC, LMR, and MHR in PCOS and healthy subjects, with or without CO. Establishing relationships between complete blood count parameters, high-sensitivity C-reactive protein (hsCRP), and hormonal, lipid and glucose metabolism in PCOS. To do this, WBC, LMR, MHR, hsCRP, anthropometric, metabolic, and hormonal data were analyzed from 395 women of reproductive age, with and without, PCOS. Correlations between MHR, and dysmetabolism, hyperandrogenism, and inflammation variables were examined. No differences were found in WBC, LMR, MHR, and hsCRP between PCOS and controls (*p* > 0.05). PCOS subjects with CO had higher hsCRP, MHR, and WBC, and lower LMR vs. those without CO (*p* < 0.05). WBC and MHR were also higher in controls with CO vs. without CO (*p* < 0.001). MHR correlated with anthropometric, metabolic, and endocrine parameters in PCOS. WHtR appeared to strongly predict MHR in PCOS. We conclude that PCOS does not independently influence WBC or MHR when matched for CO. CO and dysmetabolism may modify MHR in PCOS and control groups.

## 1. Introduction

Polycystic ovary syndrome (PCOS) is a complex metabolic disorder that occurs in 15–20% of women of reproductive age [1]. Clinical and/or biochemical hyperandrogenism, oligoovulation, and polycystic ovary morphology in ultrasound examination constitute the key factors of syndrome symptomatology, according to the 2003 Rotterdam criteria [2]. Clinical implications of PCOS include numerous endocrine, metabolic and reproductive disorders [3]. Central obesity, insulin resistance (IR), and lipid metabolism disturbances are associated with complications often observed in PCOS. We have shown previously, what is more difficult, additionally often associated with impaired dietary behaviors and knowledge of PCOS women often related to lifestyle [4,5]. PCOS is considered to be a risk factor for type 2 diabetes mellitus (DM2), metabolic syndrome (MS), accelerated atherosclerosis, and cardiovascular diseases [6,7,8]. Central obesity aggravates endocrine and metabolic disorders in PCOS [9,10].

The waist-to-height ratio (WHtR) seems to be an anthropometric index showing the highest predictive value for metabolic risk in PCOS and healthy women [11,12,13]. A WHtR of over 0.5 is a well-estimated universal cutoff for central adiposity in adults [14]. The increased WHtR could predict IR and MS. WHtR appears to be a more accurate parameter to define central adiposity and metabolic risk than the commonly used body mass index (BMI), waist circumference (WC), or waist-to-hip ratio (WHR) [13,15].

Low-grade chronic systemic inflammation is emerging as an important factor of PCOS [16]. The role of central adiposity, hyperandrogenemia, and IR in the development of chronic inflammation has been proposed [17]. Various inflammatory markers, such as high sensitive C-reactive protein (hsCRP), tumor necrosis factor α (TNFα), interleukin-1a (IL-1a), interleukin-1b (IL-1b), interleukin-6 (IL-6) and interleukin-18 (IL-18) were studied in PCOS patients [18]. Recently, inflammatory parameters that can be obtained from blood smear are gaining importance in various metabolic diseases, including PCOS [19,20,21].

Increased white blood cell (WBC) count is a recognizable risk factor for atherosclerosis in adults [22,23]. Elevated WBC count seems to accompany obesity, IR, and MS in the general population and PCOS women [24,25]. The monocyte to high-density lipoprotein cholesterol ratio (MHR) was reported as a useful indicator of systemic inflammation in patients at higher risk of MS and cardiovascular diseases [26,27,28]. PCOS was also reported to be linked with elevated peripheral lymphocyte and monocyte ratios [20,29]. Contrary findings were reported by Keskin Kurt et al., who observed low blood lymphocyte counts in PCOS women [30]. No data on the lymphocyte-to-monocyte ratio (LMR in PCOS are available in the literature.

## 2. Experimental Section

### 2.1. Materials and Methods

A total of 395 women of reproductive age (18–40 y.o.) with (*n* = 270) and without (*n* = 125) PCOS were recruited to the study at the Department of Endocrinology, Metabolism, and Internal Diseases at Poznan University of Medical Sciences, between September 2016 and February 2019. PCOS was diagnosed according to Rotterdam criteria, defined by the presence of at least two out of the following three features: Oligoovulation or anovulation, clinical and/or biochemical signs of hyperandrogenism or polycystic ovaries using ultrasound [2]. Hyperandrogenemia was defined as testosterone >2.67 nmol/L, free testosterone >11 pmol/L, and/or free testosterone index (FTI): >5.5 [2,31]. Transvaginal ultrasonography was done by a single observer. The volume and the morphology of each ovary, setting the threshold at 10 cm^3^ for increased ovarian volume and 12 for the 2–9 mm follicles, were identified [32]. Patients with extreme obesity (BMI > 40 kg/m^2^), hypertension or diagnosed heart defect, decompensated thyroid dysfunction, severe acute or chronic renal or liver diseases, Cushing’s disease, and those who were administered agents, such as birth control pills, hormonal replacement therapy, ovulation-inducing agents, and anti-androgens, over the last three months prior to the study were excluded from the study. 

Anthropometrical and clinical examination included measurement of body weight (kg), height (cm), waist circumference (WC, cm), and hip circumference (HC, cm), as well as modified Ferriman-Gallwey (mFG) scoring. mFG score of ≥ 8 was considered as hirsutism. Acne was noted as present or absent. WC was measured at the end of normal expiration midway between the lowest ribs and the iliac crest, using a non-elastic tape, according to the World Health Organization and International Diabetes Federation [33]. Waist-to-height ratio (WHtR) was used to assess central obesity. WHtR was calculated by dividing WC by height [14]. Waist-to-hip ratio (WHR) was also deliberated by dividing WC by HC. BMI was calculated as weight (kg) divided by height squared (m^2^). 

PCOS and control (CON) women were WHtR-matched. Based on their WHtR, PCOS women were divided into two groups: with central obesity—H-WHtR (WHtR ≥ 0.5, *n* = 92) and without central obesity—L-WHtR (WHtR < 0.5, *n* = 178). In addition, two WHtR-matched CON groups were formed according the same rule—with H-WHtR (WHtR ≥ 0.5, *n* = 32) and with low-WHtR (WHtR < 0.5, *n* = 93).

Informed and written consent was obtained from all participants. The clinical examination protocol complies with the Declaration of Helsinki for Human and Animal Rights and its later amendments and has received ethical approval by the Board of Bioethics of Poznan University of Medical Science.

### 2.2. Laboratory Tests

Blood samples for biochemical analyses were collected from all participants in the morning between 08:00 a.m. and 09:00 a.m. after overnight fasting. An oral glucose tolerance test (OGTT) was performed with 75 g glucose at the same day. 

A complete blood count, including WBC, monocyte, and lymphocyte counts, was assessed using a Sysmex XN-1000 hematology analyzer (Sysmex Corporation, Kobe, Japan).

Glucose measurements were performed in serum by the hexokinase method (Roche Diagnostics) with a coefficient of variation (CV) of ≤ 3%. High sensitivity C-reactive protein (hsCRP), insulin, follicle-stimulating hormone (FSH), luteinizing hormone (LH), dehydroepiandrosterone sulfate (DHEAS), estradiol (E2), total testosterone (T), sex-hormone binding globulin (SHBG), and Anti*-*Müllerian hormone (AMH) measurements were performed with the Cobas 6000 equipment (Roche Diagnostics, GmbH, Mannheim, Germany), using kits provided by the manufacturer. Lower detection limits of hormones assessed by electrochemiluminescence immunoassay (ECLIA) were presented in the Appendix A. Androstenedione concentrations were measured by the chemiluminescence method (CLIA) (Liaison XL, DiaSorin Inc. USA; a lower detection limit of 0.24 ng/mL). Free testosterone index (FTI) was determined by the following calculation (FTI) = 100 × (total testosterone/SHBG). 

Total cholesterol (TC-C), high-density lipoprotein cholesterol (HDL-C), and triglycerides (TG-C) were evaluated by the enzymatic colorimetric method. Low-density lipoprotein cholesterol (LDL-C) was estimated by the Friedewald formula: LDL-C-C = total cholesterol − HDL-C – VLDL-C (Triglycerides/5) [34]. 

The diagnosis of IR was established using the homeostasis model assessment for insulin resistance (HOMA-IR). The calculation formula for HOMA-IR was as follows: HOMA-IR = (fasting plasma glucose (mg/dL) × fasting plasma insulin (mU/L))/405 [35]. HOMA-IR > 2.5 was used as the threshold to determine IR [36].

### 2.3. Statistical Analysis

The Clinical Calculator (ClinCalc, LLC) was used to calculate the sample size [37]. The calculation was based on means and standard deviations of the WHtR database from a previous survey covering 122 PCOS women [5]. Means of the WHtR were set as 0.50 ± 0.08 (PCOS) and 0.45 ± 0.06 (CON). The minimum number of subjects for adequate study power was calculated as 30 for each independent group (H-WHtR PCOS vs. H-WHtR CON and L-WHtR PCOS vs. L-WHtR CON) with the enrolment ratio set at 1, type I error at 0.05, and power 90%. The normality of data distribution was verified with the Shapiro–Wilk test.

Differences in anthropometric, biochemical, and inflammatory parameters were calculated using the t-test of two independent samples. For non-normal distribution data, the Mann–Whitney U test was used. Descriptive statistics for quantitative variables were presented with median and interquartile range. Pearson’s linear correlation coefficients were used to explore the association between WBC, lymphocyte, monocyte counts, LMR, MHR, and metabolic and hormonal parameters. A *p* value of <0.05 was considered to be statistically significant, and all of the above-mentioned analyses were conducted using the Statistica v.13.1 statistical software (StatSoft Polska Sp. z o.o, Kraków, Poland).

A multiple stepwise forward regression was performed with MHR as the dependent variable and several selected anthropometric, endocrine, and metabolic variables as predictor variables in either PCOS and CON groups using the Stata 15.1 statistical software. 

All subjects gave their informed consent for inclusion before they participated in the study. The study was conducted in accordance with the Declaration of Helsinki, and the protocol was approved by the Ethics Committee of Poznan University of Medical Science (552/16 and 986/17).

Complete blood count inflammation markers have been intensively analyzed in PCOS, obesity, and other groups with high risk for MS. It remains unclear whether levels of inflammatory markers from hematologic indices are linked to PCOS itself, to central adiposity, or to other metabolic or hormonal disturbances.

The purpose of the present study was to compare PCOS patients with and without central obesity WHtR-matched CON with regard to levels of WBC, LMR, and MHR. The role of selected hematologic indices as possible markers of low-grade inflammation in the context of PCOS, central adiposity, and hyperandrogenemia was assessed.

## 3. Results

### 3.1. Basic Characteristic of the Study groups

Baseline anthropometrical data of the PCOS and CON groups were summarized in Table 1. Comparisons of hormonal and biochemical parameters between L-WHtR and H-WHtR patients with PCOS and L-WHtR vs. H-WHtR CON were presented in Table 2; Table 3. The assessment of anthropometric, biochemical and inflammatory parameters in whole PCOS and CON groups can be find in the Appendix A.

A total of 92 PCOS (34.1%) and 32 (25.6%) CON women enrolled in the study presented increased cardiovascular risk and had WHtR ≥ 0.5 (*p* < 0.001). The L-WHtR PCOS patients had significantly higher BMI than L-WHtR CON (*p* < 0.001). Both L-WHtR and H-WHtR PCOS patients had higher levels of LH, T, FTI, and AMH in comparison with WHtR-matched healthy CON women. The H-WHtR PCOS patients had higher levels of glucose, insulin, HOMA-IR, TC-C, LDL-C, TG-C, DHEAS, FTI, and lower levels of HDL-C and AMH than L-WHtR PCOS subjects (Table 2 and Table 3). 

The clinical characteristics of studied PCOS patients are presented in Table 4. The comparison of anthropometric, biochemical, and inflammatory parameters in the whole PCOS and CON groups was shown in the Appendix A.

### 3.2. Complete Blood Count Parameters

Parameters, such as WBC, LMR, MHR, and hsCRP, did not significantly differ between the WHtR-matched PCOS and CON groups, respectively (*p* > 0.05; Table 5). However, hsCRP, WBC, and MHR were significantly higher in H-WHtR PCOS subjects in comparison to L-WHtR PCOS patients. Likewise, WBC and MHR were significantly higher in H-WHtR CON vs. L-WHtR CON (*p* < 0.05). LMR values were higher in L-WHtR PCOS patients vs. H-WHtR PCOS subjects (*p* < 0.05, nonparametric Mann–Whitney test).

The Pearson’s correlation analysis in the PCOS group revealed that both WBC and MHR were significantly correlated with many anthropometric indices of general and central obesity, selected glucose and lipid metabolism parameters, androgens and SHBG levels (*p* < 0.05, nonparametric Spearman monotonic correlation test; Table 6). Moreover, MHR was negatively correlated to age, AMH, and FSH in PCOS patients and with FSH in the CON group. Negative associations were reported between LMR and WHtR, DHEAS, T/SHBG, FTI in PCOS patients (*p* < 0.05, nonparametric Spearman monotonic correlation test) (Table 6). Furthermore, it was observed that none of the analyzed inflammatory markers from hematologic indices was correlated with ovary volume, follicle number per ovary, acne, or mFG in PCOS (*p* > 0.05). Significantly weaker correlations between WBC, LMR, and MHR and anthropometric, metabolic, and endocrine parameters were found in the CON group (Table 6). There was also no correlation between androgens and MHR in the CON group.

In PCOS and CON patients, multiple stepwise forward regression, including MHR as the criterion variable and other selected predictors provided significant models in which the AMH, WHtR, WBC, LMR, hsCRP, TC-C, TG-C, HOMA-IR, DHEAS, FTI were retained (model in PCOS: adjusted R^2^ = 0.685, F = 46.931, *p* < 0.001; model in CON: Adjusted R^2^ = 0.600, F *=* 5.203, *p* < 0.001). WHtR, WBC, LMR, TC-C, TG-C were the strongest significant predictors of MHR in PCOS. In CON, WBC and LMR were the most important predictors of MHR (Table 7 and Table 8).

## 4. Discussion

PCOS is a multifaceted metabolic disease, linked with chronic inflammation and increased risk of MS and biochemical and/or clinical hyperandrogenism [38]. The incidence of central obesity in PCOS is typically higher than in healthy women [39,40]. The body fat distribution in PCOS women compared with weight and age-matched CON is different, due to a preponderant accumulation of visceral fat [41]. These observations are consistent with the current findings. The incidence of central adiposity, expressed as increased WHtR, was higher in PCOS than in CON (WHtR ≥ 0.5: 34.1% vs. 25.6%, *p* < 0.001).

The role of central adiposity in the development of PCOS and its complications is still under investigation [39,42]. The usefulness of WHtR as a non-invasive, universal, inexpensive, and sensitive measure is not well-established in PCOS. WHtR appears to be a good predictor of IR and MS in both PCOS and healthy women [12]. WHtR > 0.5 could be a predictor of MS in PCOS, according to Techatraisak et al. [43]. WHtR seems to be superior to other anthropometric parameters as an indicator of fat mass and systemic inflammation [44,45]. Therefore, WHtR was used as the central obesity indicator in the present study.

The main findings of this study were as follows: (1) WBC, LMR, and MHR were comparable between PCOS patients and WHtR-matched CON; (2) PCOS patients with central obesity had increased MHR levels as compared to PCOS without central obesity; similar, MHR values were higher in H-WHtR vs. L-WHtR CON; (3) LMR values were higher in L-WHtR PCOS patients vs. H-WHtR PCOS subjects (4) MHR was positively correlated with parameters of central and general adiposity, glucose, and lipid metabolism and androgen levels in PCOS.

There is little information on the link between hematological inflammation indices and WHtR in general and PCOS women. Most studies employed WHR or BMI, not WHtR. Higher WBC and MHR were found in PCOS patients with central obesity and controls (vs. PCOS patients without central obesity and controls, respectively). WBC and MHR were positively correlated with anthropometric measurements of general and central obesity in PCOS in the current study. The current data are in agreement with the results of Zhang et al., who observed a positive association between WHtR and WBC and hsCRP (r = 0.210 and 0.013, *p* < 0.001) in Chinese adults [46]. To the best of our knowledge, this is the first study showing the association between the LMR, MHR, and visceral obesity measured with WHtR in PCOS and healthy subjects.

Inconsistent data are available on the potential impact of PCOS on WBC levels. In the current study, no differences in WBC were observed between PCOS and WHtR-matched volunteers. Similarly, Shi et al., Sifler et al., and Tola et al. found no difference in WBC levels between BMI-matched PCOS patients and controls [20,47,48]. On the contrary, the elevation of WBC in PCOS subjects was observed in several other studies [19,49,50]. Numerous correlations between WBC and anthropometric parameters, androgens, and dysmetabolic parameters in the present and previous studies suggest that WBC levels could be impaired by several factors [19,48,50,51]. Simple and central obesity do not always coexist and are not the same disorder, so they could be linked with other consequences and degrees of inflammation [39].

LMR was previously studied as a determining factor in the prognosis of patients in a range of clinical situations, especially hematologic malignancies or solid neoplasm, but also heart failure and cardiac and cerebrovascular events [52,53,54]. The data suggest that a decrease in LMR is linked with a higher risk of myocardial infarction and mortality [52,53,54]. Visceral obesity appears to be linked with decreased LMR in PCOS. PCOS subjects with central obesity were characterized by lower LMR values (vs. non-obese PCOS patients) in the current study. No such association was observed in the CON group. LMR negatively correlated with WHtR and selected parameters of androgen status in PCOS women, but not in the CON. Moreover, LMR is a significant predictor of the MHR level. As far as it is known, the current study is the first to analyze LMR in PCOS. LMR appears to be an interesting inflammation factor that should be further investigated in PCOS, taking into account visceral adiposity and hyperandrogenemia.

Recently introduced, MHR constitutes the promising predictor of inflammation, thrombosis, atherosclerosis, MS, and cardiovascular events in the general population and in patients with hypertension and cardiometabolic risk. Low levels of HDL-cholesterol leads to an increased number of monocytes and their recruitment to the arterial wall. This subsequently leads (through many mechanisms) to atherosclerotic plaque formation and the release of inflammatory factors [26,27,28]. Usta et al. studied MHR values in 61 women with PCOS and 63 age- and BMI-matched healthy controls. A BMI cutoff of 25 kg/m^2^ was used to divide groups in their study. They found that MHR levels were significantly higher in PCOS vs. healthy volunteers (*p* = 0.0018). In the regression analysis, BMI, HOMA-IR, and the hsCRP constituted confounding factors modifying MHR levels. High MHR was also suggested to be an independent predictor of PCOS presence [21].

Further, Dincgez Cakmak et al. suggested MHR as a novel marker of MS in PCOS [55]. In that study, MHR levels were measured in 71 PCOS subjects (including 35 MS positive and 37 MS negative patients) and 40 healthy women, not matched for central or abdominal obesity. PCOS patients had higher MHR than CON (9.59 ± 2.82 vs. 8.2 ± 2.46, *p* < 007). The MS positive PCOS subjects also had higher MHR levels than MS negative PCOS patients (10.47 ± 2.81 vs. 8.77 ± 2.61, *p* < 0.01). Multivariate regression analysis showed that MHR was an independent predictor of MS in PCOS (>9.9, OR 3.42, 95%CI 1.41–5.78, *p* < 0.008) [55]. In the current study, no difference in MHR between WHtR-matched PCOS patients and CON was found, but there were differences between obese versus non-obese PCOS patients and between obese versus non-obese CON.

The current study used multiple stepwise forward regression to evaluate the predictive effects of selected anthropometric, androgen, inflammatory, and dysmetabolic variables on MHR levels as a dependent variable. The analysis demonstrated that WHtR, WBC, LMR, TC-C, TG-C parameters were the strongest predictors of MHR levels in PCOS. Models with retained AMH, WHtR, WBC, LMR, hsCRP, TC-C, TG-C, HOMA-IR, DHEA-S, FTI explained over 68% of MHR variance in PCOS. WHtR, WBC, LMR, TC-C, TG-C were the strongest, significant predictors of MHR in PCOS. In CON, WBC and LMR were the most important predictors of MHR.

MHR appears to be a promising predictor for central adiposity and systemic inflammation, both in PCOS and healthy women. Previous studies investigating MHR in PCOS have been potentially confounded by not adequately accounting for visceral obesity.

Moreover, neither Usta et al. nor Dincgez Cakmak et al. analyzed AMH, a significant marker of PCOS, in the context of MHR values in PCOS [56,57]. In the current study, a negative correlation between MHR and AMH was demonstrated, which is inconsistent with the hypothesis of Usta et al. on MHR as an independent marker of PCOS itself. Henes et al. reported that decreased levels of AMH were associated with chronic inflammatory diseases like rheumatoid arthritis [58]. To date, there has been no detailed available study regarding the connections between inflammatory status and AMH in patients with PCOS.

The role of hyperandrogenism in the development of systemic inflammation in PCOS is still unclear [51]. It remains uncertain whether hyperandrogenemia is a precursor of inflammation or its consequence [59]. A simple correlation analysis showed positive associations between MHR and DHEAS, T and FTI in PCOS patients, but not in the CON in the current study. Mentioned androgens appeared to slightly modify MHR levels only in case of PCOS presence. Such observations are probably the result of the frequent coexistence of visceral obesity, hyperandrogenism, and metabolic disturbances in PCOS women. Further studies on the impact of hyperandrogenism on systemic inflammation parameters from hematological indices, especially MHR, are needed.

There are some limitations to this study. The employment of the Rotterdam definition of PCOS might introduce heterogeneity in the studied PCOS patients. The comparison between PCOS patients with central obesity and WHtR-matched CON showed that H-WHtR PCOS subjects were slightly younger than control women. However, no reliable data on the link between age and MHR level were found in the publications. Further studies employing visceral adipose tissue (VAT) assessment are needed to establish the link between markers of inflammation and visceral adiposity in PCOS.

## 5. Conclusions

The current data suggest that PCOS has no independent effect on WBC or MHR when matched for abdominal obesity. MHR and WBC are disturbed in abdominal obesity, in both PCOS and eumenorrheic women. However, MHR may be a promising inflammatory and dysmetabolic marker in PCOS. The effects of central obesity on the development of systemic inflammation in PCOS should be further studied.

## Figures and Tables

**Table 1 jcm-09-03024-t001:** Baseline anthropometric findings in the polycystic ovary syndrome (PCOS) patients and control (CON) groups.

	H-WHtR PCOS (Group A) *n* = 92	H-WHtR CON (Group B) *n* = 32	*p* Value (A/B)	L-WHtR PCOS WHtR (Group C) *n* = 178	L-WHtR CON (Group D) *n* = 93	*p* Value (C/D)	*p* Value (A/C)	*p* Value (B/D)
Age	24.88	28.63	0.04	24.46	24.92	NS	NS	NS
	(22.00–30.75)	(24.33–31.87)		(21.50–27.50)	(22.67–29.00)			
Weight (kg)	83.00	79.50	NS	61.00	59.00	NS	<0.001	<0.001
	(76.50–96.00)	(71.50–88.25)		(55.50–67.00)	(54.50–64.00)			
Height (cm)	165.00	163.00	NS	168.00	168.00	NS	0.003	0.005
	(161.00–170.00)	(160.75–170.0)		(164.00–172.00)	(165.00–171.00)			
BMI (kg/m^2^)	30.86	28.84	NS	21.80	20.90	0.014	<0.001	<0.001
	(27.93–34.18)	(26.42–31.83)		(19.94–23.53)	(19.49–22.57)			
WC (cm)	95.00	89.50	NS	71.00	71.00	NS	<0.001	<0.001
	(88.00–100.00)	(84.50–97.00)		(68.00–77.00)	(68.00–74.00)			
HC (cm)	102.00	98.50	NS	84.00	82.00	NS	<0.001	<0.001
	(96.00–108.00)	(95.00–107.50)		(80.00–88.00)	(80.00–88.00)			
WHR (–)	0.93	0.92	NS	0.86	0.86	NS	<0.001	<0.001
	(0.88–0.97)	(0.90–0.96)		(0.81–0.91)	(0.82–0.90)			
WHtR (–)	0.56	0.55	NS	0.43	0.42	NS	<0.001	<0.001
	(0.53–0.61)	(0.51–0.60)		(0.41–0.46)	(0.40–0.45)			

BMI—body mass index; HC—hip circumference; H-WHtR - high waist to height ratio (with central obesity); L-WHtR - low waist to height ratio (without central obesity); WC—waist circumference; WHR—waist to hip ratio; WHtR—waist to height ratio. Data are presented as median ± interquartile range. There was used nonparametric Mann–Whitney U test. *p* < 0.05 was considered to be statistically significant; NS—not statistically significant.

**Table 2 jcm-09-03024-t002:** Biochemical parameters in the PCOS and CON groups.

	H-WHtR PCOS (Group A) *n* = 92	H-WHtR CON (Group B) *n* = 32	*p* Value (A/B)	L-WHtR PCOS (Group C) *n* = 178	L-WHtR CON (Group D) *n* = 93	*p* Value (C/D)	*p* Value (A/C)	*p* Value (B/D)
Glucose 0′ (mg/dL)	90.00	90.50	NS	86.00	87.00	NS	<0.001	<0.001
	(86.00–95.50)	(87.00–97.00)		(82.00–90.00)	(82.0–90.0)			
Insulin 0′ (mIU/mL)	15.91	12.73	NS	7.70	8.09	NS	<0.001	0.001
	(11.05–20.73)	(7.23–16.27)		(5.71–10.39)	(6.09–10.22)			
HOMA-IR (–)	3.24	3.11	NS	1.59	1.75	NS	<0.001	0.001
	(2.35–4.74)	(1.60–3.54)		(1.16–2.19)	(1.26–2.17)			
Glucose 120′ (mg/dL)	108.00	102.00	NS	90.00	93.00	NS	<0.001	NS
	(92.00–122.00)	(92.00–113.00)		(79.00–100.00)	(79.00–108.00)			
Insulin 120′ (µIU/mL)	58.44	47.50	NS	36.05	33.87	NS	<0.001	NS
	(37.05–113.00)	(28.76–67.25)		(5.71–10.40)	(24.25–50.43)			
TC (mg/dL)	183.50	179.00	NS	172.00	168.00	NS	0.045	NS
	(165.00–206.50)	(164.0–195.0)		(157.0–185.0)	(150.0–189.0)			
LDL-C (mg/dL)	106.45	101.5	NS	86.70	81.40	NS	<0.001	<0.001
	(83.75–125.80)	(87.90–119.00)		(71.20–100.30)	(69.60–95.10)			
TG-C (mg/dL)	98.50	95.00	NS	64.50	68.00	NS	<0.001	<0.001
	(72.50–149.00)	(77.0–120.0)		(49.50–83.00)	(55.00–88.50)			
HDL-C (mg/dL)	55.00	56.00	NS	69.00	73.00	NS	<0.001	<0.001
	(45.50–62.00)	(48.00–66.00)		(60.00–79.00)	(62.50–83.00)			

HDL-C—high-density lipoprotein cholesterol; HOMA-IR—homeostasis model assessment for insulin resistance index; LDL-C—low-density lipoprotein cholesterol; TC-C—total cholesterol; TG-C—triglycerides. Data are presented as median ± interquartile range. There was used nonparametric Mann–Whitney U test. *p* < 0.05 was considered to be statistically significant; NS—not statistically significant.

**Table 3 jcm-09-03024-t003:** Hormonal parameters in the PCOS and CON groups.

	H-WHtR PCOS (Group A) *n* = 92	H-WHtR CON (Group B) *n* = 32	*p* Value (A/B)	L-WHtR PCOS (Group C) *n* = 178	L-WHtR CON (Group D) *n* = 93	*p* Value (C/D)	*p* Value (A/C)	*p* Value (B/D)
FSH (mIU/mL)	6.00	5.25	NS	5.80	5.70	NS	NS	NS
	(4.80–6.90)	(4.20–6.40)		(4.60–6.80)	(4.00–7.70)			
LH (mIU/mL)	8.30	5.50	0.009	9.30	6.40	0.006	NS	NS
	(5.90–14.20)	(3.50–7.10)		(6.00–15.10)	(3.90–10.60)			
DHEAS (µg/dL)	327.50	293.00	NS	275.00	244.00	NS	0.002	NS
	(254.00–423.00)	(203.00–379.00)		(202.00–360.0)	(172.00–323.00)			
E2 (pg/mL)	37.00	63.00	0.037	44.00	46.50	NS	NS	NS
	(27.50–57.50)	(39.00–135.00)		(29.00–70.00)	(22.00–73.00)			
T (nmol/L)	1.70	1.30	0.010	1.60	1.10	<0.001	NS	NS
	(1.40–2.30)	(0.90–1.60)		(1.20–2.25)	(0.75–1.60)			
SHBG (nmol/L)	34.10	41.45	NS	64.50	74.65	NS	<0.001	<0.001
	(25.90–51.30)	(31.60–58.10)		(49.5–91.60)	(52.25–159.75)			
FTI (%)	4.74	3.35	0.026	2.54	1.55	<0.001	<0.001	0.004
	(3.08–7.46)	(1.72–4.48)		(1.77–4.00)	(0.98–2.74)			
A (ng/mL)	4.01	3.27	0.048	3.99	3.04	0.005	0.523	NS
	(2.96–5.02)	(2.29–4.10)		(2.95–5.30)	(2.27–4.39)			
AMH (pmol/L)	44.58	16.42	<0.001	57.05	28.06	<0.001	<0.001	0.014
	(36.45–61.73)	(13.12–24.09)		(41.79–85.63)	(24.78–36.34)			

A—androstenedione; AMH—Anti-Müllerian hormone; DHEAS—dehydroepiandrosterone sulfate; E2—estradiol; FSH—follicle-stimulating hormone; FTI—free testosterone index; LH—luteinizing hormone; SHBG—sex hormone binding globulin; T—total testosterone. Data are presented as median ± interquartile range. There was used nonparametric Mann–Whitney U test. *p* < 0.05 was considered to be statistically significant; NS—not statistically significant.

**Table 4 jcm-09-03024-t004:** Clinical characteristics of studied PCOS patients.

Criterion	All PCOS *n* = 270	H-WHtR PCOS *n* = 92	L-WHtR PCOS *n* = 178
Acne	155 (57.4%)	48 (52.2%)	107 (60.1%)
Hirsutism	150 (55.6%)	58 (63.0%)	92 (51.7%)
Polycystic ovaries *	211 (78.1%)	73 (79.3%)	138 (77.5%)

* volume and the morphology of each ovary, setting the threshold at 10 cm^3^ for increased ovarian volume and 12 for the 2–9 mm follicles.

**Table 5 jcm-09-03024-t005:** Inflammatory parameters for PCOS and CON groups.

	H-WHtR PCOS (Group A) *n* = 92	H-WHtR CON (Group B) *n* = 32	*p* Value (A/B)	L-WHtR PCOS (Group C) *n* = 178	L-WHtR CON (Group D) *n* = 93	*p* Value (C/D)	*p* Value (A/C)	*p* Value (B/D)
hsCRP (mg/L)	2.10	1.35	NS	0.50	0.70	NS	<0.001	NS
	(1.00–4.10)	(0.45–3.65)		(0.30–1.00)	(0.30–1.25)			
WBC × 10^3^/µL	6.98	6.84	NS	5.75	5.53	NS	<0.001	<0.001
	(5.99–8.27)	(4.87–9.86)		(4.80–6.71)	(4.74–6.80)			
MHR (–)	8.88	8.59	NS	5.85	6.34	NS	<0.001	<0.001
	(6.80–11.72)	(7.72–10.63)		(4.78–7.41)	(5.12–8.26)			
LMR (–)	4.03	4.14	NS	4.52	4.11	NS	0.014	NS
	(3.38–5.18)	(3.45–5.35)		(3.78–5.77)	(3.22–5.52)			

hsCRP—high sensitivity C-reactive protein; LMR—lymphocyte to monocyte ratio; MHR—monocyte-to-high-density lipoprotein cholesterol ratio; WBC—white blood cell count. Data are presented as median ± interquartile range. There was used nonparametric Mann–Whitney U test. *p* < 0.05 was considered to be statistically significant. NS—not statistically significant.

**Table 6 jcm-09-03024-t006:** Pearson’s linear correlation analysis of complete blood count inflammatory markers in PCOS and CON groups.

Parameter	PCOS			CON		
	WBC	LMR	MHR	WBC	LMR	MHR
	r, *p* Value	r, *p* Value	r, *p* Value	r, *p* Value	r, *p* Value	r, *p* Value
Anthropometric Parameters						
**Age**	NS	NS	−0.170; 0.045	NS	NS	NS
**Weight**	0.358; <0.001	NS	0.390; <0.001	NS	NS	0.529; 0.006
**Height**	NS	NS	NS	NS	NS	−0.405; 0.006
**WC**	0.335; <0.001	NS	0.425; <0.001	NS	NS	0.610; 0.001
**HC**	0.321; <0.001	NS	0.317; <0.001	NS	NS	0.580; 0.002
**BMI**	0.369; <0.001	NS	0.452; <0.001	NS	NS	0.630; <0.001
**WHR**	0.129; 0.035	−0.127; <0.001	0.278; <0.001	NS	NS	NS
**WHtR**	0.446; <0.001	−0.128; 0.038	0.451; <0.001	NS	NS	0.652; <0.001
Inflammation Parameters						
**WBC**		NS	0.615; <0.001		NS	0.778; <0.001
**LMR**	NS		−0.438; <0.001	NS		−0.439; 0.028
**MHR**	0.615; <0.001	−0.438; <0.001		0.778;<0.001	−0.439; 0.028	
**hsCRP**	0.373; <0.001	NS	0.399; <0.001	NS	NS	NS
Biochemical Parameters						
**TC-C**	NS	NS	−0.171; 0.007	NS	0.441; 0.027	NS
**LDL-C**	NS	NS	NS	NS	NS	NS
**TG-C**	0.328; <0.001	NS	0.436; <0.001	NS	NS	NS
**HDL-C**	−0.324; <0.001	NS	−0.699; <0.001	NS	NS	−0.738; <0.001
**Glucose 0′**	0.152; 0.011	NS	0.237; <0.001	NS	NS	NS
**glucose 120′**	0.174; 0.004	NS	0.241; <0.001	NS	NS	0.404; 0.045
**insulin 0′**	0.348; <0.001	NS	0.450; <0.001	NS	NS	0.440; 0.028
**insulin 120′**	0.258; <0.001	NS	0.320; <0.001	0.399; 0.048	NS	0.531; 0.006
**HOMA-IR**	0.342; <0.001	NS	0.463; <0.001	NS	NS	0.472; 0.017
Hormonal Parameters						
**FSH**	−0.142; 0.019	NS	−0.420; <0.001	NS	NS	−0.396; 0.049
**LH**	NS	NS	NS	NS	NS	NS
**AMH**	NS	NS	−0.213; 0.001	NS	NS	NS
**DHEAS**	0.212; <0.001	−0.183; 0.003	0.302; <0.001	NS	NS	NS
**E2**	NS	NS	NS	0.455; 0.022	NS	NS
**T**	0.130; 0.031	NS	0.178; 0.005	NS	NS	NS
**SHBG**	−0.348; <0.001	0.161; 0.008	−0.437; <0.001	NS	NS	NS
**FTI**	0.361; <0.001	−0.163; 0.008	0.445; <0.001	NS	NS	NS
**A**	NS	NS	NS	NS	NS	NS

A—androstenedione; AMH—Anti-Müllerian hormone; BMI—body mass index; DHEAS—dehydroepiandrosterone sulfate; E2—estradiol; FSH—follicle-stimulating hormone; FTI—free testosterone index; HC—hip circumference; HDL-C—high-density lipoprotein cholesterol; HOMA-IR—homeostasis model assessment for insulin resistance index; hsCRP—high sensitivity C-reactive protein; LDL-C—low-density lipoprotein cholesterol; LH—luteinizing hormone; LMR—lymphocyte to monocyte ratio; mFG—modified Ferriman-Gallwey scoring; MHR—monocyte-to-high-density lipoprotein cholesterol ratio; SHBG—sex hormone binding globulin; T—total testosterone; TC-C—total cholesterol; TG-C—triglycerides; WBC—white blood cell count. WC—waist circumference; WHR—waist to hip ratio; WHtR—waist to height ratio; NS—not statistically significant.

**Table 7 jcm-09-03024-t007:** Multiple stepwise forward regression between monocyte-to-HDL ratio (MHR) and selected predictors in PCOS and CON groups/Multivariate logistic regression analysis of factors related to/affecting monocyte-to-HDL ratio (MHR) in PCOS and CON groups.

Model	R	R^2^	Adjusted R^2^	SEE	F Change	*p* Value
1-PCOS *	0.837	0.700	0.685	1.9107	46.931	<0.001
2-CON *	0.862	0.743	0.600	2.3363	5.203	<0.001

* predictors: AMH, WHtR, WBC, LMR, hsCRP, TC-C, TG-C, HOMA-IR, DHEA-S, FTI.

**Table 8 jcm-09-03024-t008:** Stepwise linear regression model for the assessment of monocyte-to-HDL ratio (MHR) in PCOS and CON patients.

	Model 1—PCOS				Model 2—CON			
Predictor	β	Standard Error of β	t	*p* Value	β	Standard Error of β	t	*p* Value
AMH (pmol/L)	−0.050	0.042	−1.176	0.241	0.097	0.143	0.675	0.508
WHtR	0.117	0.054	2.164	0.032	0.266	0.242	1.100	0.286
WBC *×* 10^3^/µL	0.442	0.045	9.760	<0.001	0.470	0.147	3.194	0.005
LMR	−0.317	0.040	−7.960	<0.001	−0.366	0.173	−2.116	0.049
hsCRP (mg/L)	0.004	0.041	0.086	0.931	0.219	0.162	1.349	0.194
TC-C (mg/dL)	−0.205	0.043	−4.822	<0.001	−0.164	0.182	−0.905	0.378
TG-C (md/dL)	0.294	0.049	6.126	<0.001	0.323	0.212	1.528	0.144
HOMA-IR	0.016	0.055	0.286	0.775	−0.014	0.302	−0.045	0.965
DHEA-S (µg/dL)	0.002	0.049	0.051	0.960	0.139	0.181	0.768	0.452
FTI (%)	0.077	0.057	1.364	0.174	−0.273	0.242	−1.127	0.274

AMH—Anti-Müllerian hormone; *β -* the standarized beta; DHEAS—dehydroepiandrosterone sulfate; FTI—free testosterone index; HOMA-IR—homeostasis model assessment for insulin resistance index; hsCRP—high sensitivity C-reactive protein; LMR—lymphocyte to monocyte ratio; SEE - standard error of estimate; TC-C—total cholesterol; TG-C—triglycerides; WBC—white blood cell count. WHtR—waist to height ratio.

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
