# Peer review of "Effect of Central Obesity and Hyperandrogenism on Selected Inflammatory Markers in Patients with PCOS: A WHtR-Matched Case-Control Study"

_jcm, 2020, doi:10.3390/jcm9093024_

Round 1
Reviewer 1 Report
Kaluzna et al present a prospective study about the effects of hyperandrogenism and selected inflammatory markers in patients with PCOS. As Inflammation represents an important issue in patients with PCOS, the topic is appealing, however, the manuscript lacks some important Information and needs to be improved.
Major issues:
- In section 2.3, power calculation is provided, but insufficiently. It is not clear if 67 for each independent group is ment for PCOS vs. controls or within both groups for L-WHtR vs L-WHtR. Moreover, the number of controls with H-WHtR is much smaller than that of PCOS with H-WHtR (around 1/3).
- Please provide a comparison of all PCOS patients vs all controls. Results are mostly presented in subgroups based on WHtR. Especially since in table 2, correlations seem to be provided for all patients per group.
- The title is somehow misleading. The study is around differences in inflammatory markers in subgroups based on the WHtR, equaling central obesity. Possible influence or effecht of hyperandrogenism is, however, neglected. Also, androstenedione is commonly measured for PCOS, which is missing here. How many patients suffer from hyperandrogenism (acne, hirsutism, etc.) - numbers not provided.
- Please clarify on what basis you have chosen These markers on long-term inflammation as there are others available too (neutrophil-to-lymphocyte-ratio, NLR; platelet-to-lymphocyte-ratio, PLR; etc).
- It is not clear how resuts in tables are presented (median/mean, min-max or IQR, etc.)
- In the methods part (section 2.1), diagnosis of PCOS is described. Free Testosterone is mentioned, but results are missing. Moreover, hyperandrogenemia seems to be missing or only pressent in few patients with PCOS?
testosterone (PCOS >2.67): 1.70 (1.40-2.30)
FTI (PCOS >5.5): 4.74 (3.08-7-46) - In Results (3.1), it is mentioned that "all analyzed inflammatory markers were correlated with ovary volume, follicle number per ovary, acne or mFG in PCOS" but results are not shown. it is also not given how many patients display acne, hirsutism, polycystic ovarian morphology. again, it is not clear if this correlation refers to all patients or only a subgroup with low or high central obesity.
- abstract, line 31: WBC and MHR were higher in controls with and without central obesity (<0.005). This is in contrast to results, please clarify.
Minor issues:
- be consisted with the terms used. For example, in the text you use L/H-WHtR, but in the tables it is written differently.
- Table 1: in the first line, "group C" is missing underneath PCOS WHtR<0,5.
Author Response
Thank you very much for your thorough review of the manuscript and very pertinent comments. We hope that the amendments improved the text and made the work more clear.
Major issues:
- In section 2.3, power calculation is provided, but insufficiently. It is not clear if 67 for each independent group is ment for PCOS vs. controls or within both groups for L-WHtR vs L-WHtR. Moreover, the number of controls with H-WHtR is much smaller than that of PCOS with H-WHtR (around 1/3).
Thank you, the data on power calculation were added and clarified in the text.
- Please provide a comparison of all PCOS patients vs all controls. Results are mostly presented in subgroups based on WHtR. Especially since in table 2, correlations seem to be provided for all patients per group.
Thank you for your advice, a comparison of all PCOS patients vs all controls was provided in Supplementary materials. Correlations were provided for all patients per group in order not to obscure text.
- The title is somehow misleading. The study is around differences in inflammatory markers in subgroups based on the WHtR, equaling central obesity. Possible influence or effecht of hyperandrogenism is, however, neglected. Also, androstenedione is commonly measured for PCOS, which is missing here. How many patients suffer from hyperandrogenism (acne, hirsutism, etc.) - numbers not provided.
Thank you for your advice. The data on the clinical characteristics of studied PCOS patients was added to the characteristic of the patients (section 3.1; Table 2). The PCOS women most often presented polycystic ovaries at ultrasound, followed by acne (57.4% in whole PCOS group; more popular in the L-WHtR PCOS group) and hirsutism (55.6% in entire PCOS group; more popular in the H-WHtR PCOS group). Data on androstendione level was also added to the text and Table 1c and 3. We focused on possible influence of biochemical hyperandrogenism on inflammatory markers, because we belived that assessing androgen levels is more objective than potential clinical signs such as hirsutism and acne. Both hirsutism and acne may have causes other than biochemical hyperandrogenism. Acne can be found quite often and hirsutism occasionally in healthy women (1). There was no correlation between androstendione levels and selected inflammation indices.
Please clarify on what basis you have chosen These markers on long-term inflammation as there are others available too (neutrophil-to-lymphocyte-ratio, NLR; platelet-to-lymphocyte-ratio, PLR; etc).
Many hematological inflammation indices were discussed so far in the literature in context of PCOS. We chose the newest and the least discussed so far inflammation markes in PCOS (no data on lymphocyte-to-monocyte ratio in PCOS so far; only 2 reports on monocyte-to-high-density lipoprotein cholesterol ratio (MHR) levels in PCOS (2, 3)). White blood cell count (WBC) was chose as the most classic, powerful inflammatory indice, boradly discussed in PCOS and other metabolic diseases. We found the most interesting observations and correlations between exacly MHR, LMR and WBC and antropometric, metabolic and hormonal parameters in our material in our opinion. The connection between central adiposity and levels of neutrophil-to-lymphocyte-ratio, NLR; platelet-to-lymphocyte-ratio, PLR; etc would also be interesting and we hope to present it in future papers. In conference abstract presented in 20th European Congress of Endocrinology in Barcelona in 2018 "The inflammatory markers and central obesity in policystic ovary syndrome" (authors: Małgorzata Kałużna, Adam Janicki, Magdalena Człapka-Matyasik Katarzyna Wachowiak-Ochmańska, Jerzy Moczko, Katarzyna Ziemnicka & Marek Ruchała), we investigated and discussed apart from WBC and LMR, also platelet to white blood cell ratio (PLT/WBC) in PCOS women (https://www.endocrine-abstracts.org/ea/0056/ea0056p93).
- It is not clear how results in tables are presented (median/mean, min-max or IQR, etc.)
The sentence: "Data are presented as median± interquartile range. There was used nonparametric Mann-Whitney U test. p<0.05 was considered to be statistically significant" is below the Table 1d, and also Table 1a-c.
- In the methods part (section 2.1), diagnosis of PCOS is described. Free Testosterone is mentioned, but results are missing. Moreover, hyperandrogenemia seems to be missing or only pressent in few patients with PCOS?
testosterone (PCOS >2.67): 1.70 (1.40-2.30)
FTI (PCOS >5.5): 4.74 (3.08-7-46)
Thank you for your comment. In our study 43 PCOS women had total testosterone above 2.67 nmol/l, while 63 PCOS women had FTI above 5.5%. Clinical hyperandrogenism was more popular than biochemical hyperandrogenism in our group of patients. Part of the PCOS patients presented only acne, part only hirsutism. Data on clinical characteristic of PCOS group was presented in Table 2. The Rotterdam criteria only require two of three features: hyperandrogenism, ovulatory dysfunction, and polycystic ovarian morphology on ultrasound. So, a woman with oligomenorrhoea and policystic ovaries on ultrasound can be diagnosed as having PCOS. Moreover, hyperadrogenism may be clinical or biochemical according to the above criteria. In chapter “limitations” we mentioned the disadvantages of the employment of the Rotterdam definition of PCOS.
We hope that the amendments made the issue of hyperandogenism in our work clearer.
- In Results (3.1), it is mentioned that "all analyzed inflammatory markers were correlated with ovary volume, follicle number per ovary, acne or mFG in PCOS" but results are not shown. it is also not given how many patients display acne, hirsutism, polycystic ovarian morphology. again, it is not clear if this correlation refers to all patients or only a subgroup with low or high central obesity.
Thank you, a typographical error stuck to the sentece when a native speaker checked the text. The proper sentence is: "none of analyzed inflammatory markers was correlated with ovary volume, follicle number per ovary, acne or mFG in PCOS" The remainder of the text has been checked for such errors. The data on the clinical characteristics of studied PCOS patients was added to the characteristic of the patients (section 3.1; Table 2).
- abstract, line 31: WBC and MHR were higher in controls with and without central obesity (<0.005). This is in contrast to results, please clarify.
A typographical error stuck to the abstract when a native speaker checked the text. The remainder of the text has been checked for such errors. The proper sentence is: "WBC and MHR were also higher in controls with CO vs. without CO (p<0.001)".
- be consisted with the terms used. For example, in the text you use L/H-WHtR, but in the tables it is written differently.
Thank you for your valuable attention, the terms L/H-WHtR were unified in whole text.
- Table 1: in the first line, "group C" is missing underneath PCOS WHtR<0,5.
Thank you for your valuable advice, the term "group C" was added.
Reviewer 2 Report
This study thoroughly examines circulating inflammatory markers in PCOS and control women in comparison to a raft of other metabolic and hormonal measures. The sample size (270 PCOS and 125 control women) seems robust and was guided by clear sample size calculations. In general, PCOS did not exert independent effects on various markers of inflammation; the effects were rather due to abdominal obesity. The research is well conducted and presented and I have very few concerns.
General queries
- An indication of the sensitivity of the hormone assays should be given.
- Line 318: “A simple correlation analysis showed positive associations between MHR and DHEAS, T and FTI in PCOS patients, but not in the CON in the current study. FTI and DHEA-S appeared not to significantly modify MHR levels in PCOS or heathy women.” Has these two sentence contradictory? Has the second sentence undersold a PCOS-specific effect of hyperandrogenism on inflammation?
Specific/minor queries
- LMR is not defines, except in Table 1d.
Author Response
Thank you very much for your review of the manuscript and very applicable comments. We hope that the revision made improved the text and made the work clearer.
General queries
- An indication of the sensitivity of the hormone assays should be given.
Thank you for your advice, the indications of the sensitivity of the hormone assays were added to the section "Methods" .
- Line 318: “A simple correlation analysis showed positive associations between MHR and DHEAS, T and FTI in PCOS patients, but not in the CON in the current study. FTI and DHEA-S appeared not to significantly modify MHR levels in PCOS or heathy women.” Has these two sentence contradictory? Has the second sentence undersold a PCOS-specific effect of hyperandrogenism on inflammation?
Thank you, that is right. Unfortunately, correlations between MHR levels and chosen androgens are weak. We changed second sentence to: "Mentioned androgens appeared to slightly modify MHR levels only in case of PCOS presence".
Specific/minor queries
- LMR is not defines, except in Table 1d.
The explanation of abbreviations were added under Tabele 1 a-c, Table 2 and 3.
- Johnstone EBea. Lack of association between clinical and biochemical hyperandrogenism in patients with polycystic ovary syndrome. Fertility and Sterility. 2008;90:16.
- Usta A, Avci E, Bulbul CB, Kadi H, Adali E. The monocyte counts to HDL cholesterol ratio in obese and lean patients with polycystic ovary syndrome. Reprod Biol Endocrinol. 2018 Apr 10;16(1):34.
- Dincgez Cakmak B, Dundar B, Ketenci Gencer F, Aydin BB, Yildiz DE. TWEAK and monocyte to HDL ratio as a predictor of metabolic syndrome in patients with polycystic ovary syndrome. Gynecol Endocrinol. 2019 Jan;35(1):66-71.
Round 2
Reviewer 1 Report
The sentence: "Data are presented as median± interquartile range. There was used nonparametric Mann-Whitney U test. p<0.05 was considered to be statistically significant" is below the Table 1d, and also Table 1a-c. "
This should also be added to the material & Methods sections About statistics.